Non-avian theropod phalanges from the marine Fox Hills Formation (Maastrichtian), western South Dakota, USA

Chamberlain, Jr John A. 1 2 johnc@brooklyn.cuny.edu
Knoll Katja 3
http://orcid.org/0000-0001-8096-3605 J. W. Sertich Joseph 4 5
1 Department of Earth and Environmental Sciences and Department of Biology, City University of New York, Graduate School and University Center , New York City, New York , United States
2 Department of Earth and Environmental Sciences, City University of New York, Brooklyn College , Brooklyn, New York , United States
3 Paria River District, US Bureau of Land Management , Kanab, Utah , United States
4 Smithsonian Tropical Research Institute , Panama City , Panama
5 Department of Geosciences, Warner College of Natural Resources, Colorado State University , Fort Collins, Colorado , United States
Hutchinson John
Electronic publication date: 2023 Feb 7
Publication date: 2023
Volume: 11
Electronic Location ID: e14665
Received 2022 Apr 11; Accepted 2022 Dec 9
Copyright: © 2023 Chamberlain, Jr et al.
Copyright year: 2023
Copyright holder: Chamberlain, Jr et al.
License: This is an open access article distributed under the terms of the Creative Commons Attribution License, which permits unrestricted use, distribution, reproduction and adaptation in any medium and for any purpose provided that it is properly attributed. For attribution, the original author(s), title, publication source (PeerJ) and either DOI or URL of the article must be cited.
License URL: https://creativecommons.org/licenses/by/4.0/

Keywords: Non-avian theropod phalanges; Marine preservation; Fox Hills Formation; Maastrichtian; South Dakota, USA

Funding: PSC-CUNY Research Award Program of the City University of New York 62274, 64246, 64263, 668230 and 669224 This research was supported by the PSC-CUNY Research Award Program of the City University of New York (grant numbers 62274; 64246; 64263; 668230; and 669224). The funders had no role in study design, data collection and analysis, decision to publish, or preparation of the manuscript.

==============================
We report here the first dinosaur skeletal material described from the marine Fox Hills Formation (Maastrichtian) of western South Dakota. The find consists of two theropod pedal phalanges: one recovered from the middle part of the Fairpoint Member in Meade County, South Dakota; and the other from the Iron Lightning Member in Ziebach County, South Dakota. Comparison with pedal phalanges of other theropods suggests strongly that the Fairpoint specimen is a right pedal phalanx, possibly III-2, from a large ornithomimid. The Iron Lightning specimen we cautiously identify as an ornithomimid left pedal phalanx II-2. The Fairpoint bone comes from thinly bedded and cross-bedded marine sandstones containing large hematitic concretions and concretionary horizons. Associated fossils include osteichthyan teeth, fin spines and otoliths, and abundant teeth of common Cretaceous nearshore and pelagic chondrichthyans. Leaf impressions and other plant debris, blocks of fossilized wood, and Ophiomorpha burrows are also common. The Iron Lightning bone comes from a channel deposit composed of fine to coarse sandstone beds, some of which contain bivalves, and a disseminated assemblage of mammal teeth, chondrichthyan teeth, and fragmentary dinosaur teeth and claws. We interpret the depositional environment of the two specimens as marginal marine. The Fairpoint bone derives from a nearshore foreset setting, above wave base subject to tidal flux and storm activity. The Iron Lightning specimen comes from a topset channel infill probably related to deposition on a tidal flat or associated coastal setting. The taphonomic history and ages of the two bones differ. Orthogonal cracks in the cortical bone of the Fairpoint specimen suggest post-mortem desiccation in a dryland coastal setting prior to transport and preservation in the nearby nearshore setting described above. The pristine surface of the Iron Lightning specimen indicates little transport before incorporation into the channel deposit in which it was found. The Fairpoint bone bed most probably lies within the Hoploscaphites nicolletii Ammonite Zone of the early late Maastrichtian, and would therefore have an approximate age of 69 Ma. The Iron Lightning bone is from the overlying H. nebrascensis Ammonite Zone, and is thus about one million years younger.

Introduction

The Fox Hills Formation is a silty to sandy, fossiliferous nearshore to onshore deposit of Maastrichtian age that separates the marine shales of the Pierre Formation from the overlying terrestrial, dinosaur-rich Hell Creek and Lance formations of the late Maastrichtian. In South Dakota, the Fox Hills Formation is exposed along a sinuous outcrop belt that curves around the northern and western flanks of the Black Hills (Fig. 1). To the east of the Black Hills lie two Fox Hills outliers, separated from the main trend of Fox Hills exposures, and from each other, by erosion of the Cheyenne River and its tributaries. These outliers are referred to here as the Fairpoint area (blue in Fig. 1) and the Badlands National Park area (brown in Fig. 1). In the Fox Hills Type Area (brown in Fig. 1), and in North Dakota, the lower part of the Fox Hills Formation (Trail City and Timber Lake members) is interpreted as a wedge of marine sand and silt prograding southwestward across the western interior basin (Waage, 1961, 1968; Landman & Waage, 1993). The sandy upper unit of the Type Area Fox Hills Formation (Iron Lightning Member) and the sandy Fox Hills exposures to the west of the Type Area represent the eastward and southeastward progradation of deltaic and shoreline deposits, referred to as the Sheridan Delta in Wyoming and Montana by Gill & Cobban (1973). These patterns of sediment migration are associated with the final closing of the Western Interior Seaway during the late Maastrichtian and early Danian (Waage, 1968; Erickson, 1999; Gill & Cobban, 1973). Although terrestrial, lignitic horizons occur in the Fox Hills Formation of South Dakota (Waage, 1961, 1968; Black, 1964; Pettyjohn, 1967), this unit is primarily composed of marine sediments containing a macrofauna dominated by marine invertebrates, particularly gastropods (Erickson, 1974), bivalves (Speden, 1970; Erickson, 1978), and ammonites (Landman & Waage, 1993).

Figure 1 Locality Map showing the exposure areas of the Fox Hills Formation surrounding the Black Hills of western South Dakota, USA. We produced this map using ArcGIS software from recent geologic maps and GIS datasets in the public domain published by the state geological surveys of South Dakota (Martin et al., 2004); North Dakota (Clayton et al., 1980); Montana (Vuke et al., 2007); and Wyoming (Love & Christiansen, 1985), and available on the United States Geological Survey website. Outcrop areas of primary interest in the present paper are the Fairpoint Area (blue) studied by Pettyjohn (1967), Becker, Chamberlain & Terry (2004, 2009); the Badlands National Park Area (brown) studied by Chamberlain et al. (2001), Stoffer et al. (2001), Jannett & Terry (2008), Landman et al. (2013); and the Fox Hills Type Area (orange) studied by Waage (1961, 1968), Speden (1970), and Landman & Waage (1993). Other Fox Hills outcrop areas are shown in black. Black Star—collection site of the theropod phalanx DMNH EPV.138575. White Star—collection site of the theropod phalanx YPM VP.061075. Gray Dot in Badlands area—Fox Hills outcrop at Creighton, SD, discussed in text. White dot in eastern Wyoming—Fox Hills outcrop at Red Bird, WY, discussed in text. INSET: a detailed map of White Owl, South Dakota, showing the location of the DMNH EPV.138575 outcrop (black star) discussed here (DMNH locality number 19383).

Remains of terrestrial animals, and dinosaurs in particular, are only rarely recovered from the Fox Hills Formation, even from its terrestrial beds. Hoganson, Erickson & Holland (2007) describe small theropod tooth fragments recovered from sites in the Bullhead lithofacies of the lowermost Iron Lightning Member in southcentral North Dakota. Waage (1968, pg. 127 and again on pg. 133) mentions a similar collection of fragmentary dinosaur remains, primarily tooth and claw fragments, from a channel deposit in the Colgate lithofacies of the Iron Lightning Member in the Fox Hills Type Area of northcentral South Dakota. In his article erecting the Fairpoint and White Owl Creek members as formal units of the Fox Hills Formation in the Fairpoint area of western South Dakota, Pettyjohn (1967) mentions anecdotally that he encountered dinosaur bones in the middle part of the Fairpoint Member (“a few dinosaur and turtle bones as well as shark teeth were found throughout this unit,” Pettyjohn, 1967, pg. 1364). He did not describe this material, however. The fate of Pettyjohn’s Fox Hills dinosaur material is unknown, and as far as can be discerned, it does not appear that it was retained for future study.

In this article we describe two small theropod phalanges from marine beds of the Fox Hills Formation. The first of these is from the Fairpoint Member at White Owl in the Fairpoint area of western South Dakota (Fig. 1), recovered as a by-product of earlier work by one of us (JAC) on fossil fish occurring in these same beds (Becker, Chamberlain & Terry, 2004; Becker et al., 2009). The second is from a small assemblage of undescribed dinosaur material reported by Waage (1968) from the Iron Lightning Member in Ziebach County, South Dakota, about 190 km northeast of the Fairpoint locality (Fig. 1). Given the overall rarity of dinosaur remains in the Fox Hills Formation, particularly in western South Dakota, a formal description of these bones is warranted.

The Fairpoint specimen from White Owl, Meade County, South Dakota has been deposited in the vertebrate paleontology collections of the Denver Museum of Nature and Science (DMNH), Denver, Colorado, USA, and is identified by the catalogue number: DMNH EPV.138575. The Iron Lightning specimen from the Fox Hills Type Area in Ziebach County, South Dakota, is in the vertebrate paleontology collections of the Yale Peabody Museum, Yale University, New Haven, Connecticut, USA, and carries the catalogue number: YPM VP.061705.

Collecting localities

DMNH EPV.138575: Specimen DMNH EPV.138575 was collected in Section 35, T7N, R14E (DMNH loc. 19383), about 13 km southeast of Enning, southeastern Meade County, South Dakota (Fig. 1). This is the same site that yielded chondrichthyan teeth described in Becker, Chamberlain & Terry (2004) and osteichthyan remains detailed in Becker et al. (2009). DMNH EPV.138575 was recovered from a soft, laminated, well-sorted sandstone, white to tan in color, exposed near the top of a hillside above Pine Creek (Fig. 2). The sandy beds form pedestals arrayed beneath a dense pattern of hard, well-cemented, reddish-brown hematitic concretions, some up to 3 m in diameter. The concretions are ovoid in shape and coalesce in places, forming well defined horizons resistant to weathering and erosion. Two main concretion horizons are visible in Fig. 2: an upper one (foreground, Fig. 2); and a lower one about 3 m lower in the section (middle ground, Fig. 2). The sandstone is finely bedded, with hummocky, high-angle tangentially cross-stratified beds and trough cross bedding commonly occurring throughout the outcrop, including within concretions (Figs. 3A and 3B). Some smaller fossils, including teeth and other piscine material figured in Becker, Chamberlain & Terry (2004) and Becker et al. (2009), are scattered within the soft sandstone and sandstone pedestals. Many of the teeth are sediment polished and missing delicate crown and root elements. However, most small fossils are found weathered out of the sandstone and lying in the piles of loose sand distributed around the bases of the pedestals or in the debris mounds of harvester ant colonies located in the outcrop area (Becker, Chamberlain & Terry, 2004). DMNH EPV.138575 was found lying in situ, and parallel to bedding in the sandstone pedestal identified in Fig. 2. Ophiomorpha burrows (Fig. 3C) are found both in the soft sandstone as well as in the hard concretions. Leaf impressions (Fig. 3D) and plant debris occur within the concretions overlying the phalanx horizon, and small blocks of fossil wood are scattered throughout the loose sand at the base of the pedestals.

Figure 2 White Owl outcrop of the Fairpoint Member yielding DMNH EPV.138575.

White Owl outcrop of the Fairpoint Member looking southwesterly from the top of the hill on which the outcrop occurs. Visible are two prominent horizons of coalesced, dark colored, iron stained sandstone concretions separated from one another by several meters of buff colored, soft sandstone. DMNH EPV.138575 was recovered from a soft sandstone pedestal standing beneath the upper concretion horizon at the position marked by the white square. Photography by John Chamberlain.

Figure 3 Sedimentary features and fossils of the White Owl theropod site.

Sedimentary features and fossils of the White Owl locality where DMNH EPV.138575 was found. (A) Soft sandstone below the lower of the two concretion horizons showing trough cross bedding above laminar horizontal bedding.Scale bar = 25 cm. (B) Cross bedding in soft sandstone above the higher of the two concretion horizons. Scale bar = 1 m. (C) Ophiomorpha burrow from a hematitic concretion. Marker pen is 14 cm in length. (D) Positive and negative of leaf impression from a hematitic concretion. Scale bar = 2 cm. The specimen is probably a leaf fragment of the buckthorn, Rhamnus salicifolius, which is known from the Fox Hills Formation in North Dakota (Peppe, Erickson & Hickey, 2007). Photography by John Chamberlain.

YPM VP.061705: Waage (1968) indicates that specimen YPM VP.061705 was found in Sec. 33, T14N; R19E, Ziebach County, SD (YPM locality 74). The YPM specimen comes from Waage’s (1968) type section of the Iron Lightning Member where it was measured in the SW corner of a drainage divide in the badlands located to the east of the gravel road running northward from Highway 212 to the village of Iron Lightning near the Moreau River. Waage (1968, pg. 133) describes the Colgate lithofacies sand body containing YPM VP.061705 as a sandy, very fine to medium grained subgraywacke about 12 m thick, which weathers grayish white. Present are thin bands of iron stained shale and some carbonaceous laminae. Cross bedding is prominent in these beds. Also common are brown-colored ovoid concretions up to 4 m long. The basal portion of the unit contains fossil-rich lenses and infilled channel sandstones preserving Corbicula, Crassostrea, Anomia, and fish teeth, primarily of the guitarfish Myledaphus bipartitus. Also present are otoliths, wood fragments, mammal teeth, and fragmentary dinosaur remains. The latter consists of broken hadrosaur, ceratopsid, and theropod teeth, fragmentary theropod claws, and YPM VP.061705. In his measured Iron Lightning type section, Waage (1968, pg. 133) indicates that this channel cut dinosaur horizon in the Colgate lithofacies lies about 14 m below the base of the overlying Hell Creek Formation.

Geologic setting

DMNH EPV.138575: Pettyjohn (1967) recognized two stratigraphically distinct members in the Fox Hills Formation in the Fairpoint area of western South Dakota: the Fairpoint Member and the White Owl Creek Member. The Fairpoint Member, which overlies the Pierre Shale, is the lower of the two members. It is about 50 m thick and consists primarily of light-colored marine sands containing channel incisions, cross beds, and with occasional horizons of dark, hematitic concretions. The uppermost part of the Fairpoint Member takes on a distinctly continental character in that it contains numerous lignite beds (the Stoneville Lithofacies of Pettyjohn (1967)). The White Owl Creek Member, the upper of Pettyjohn’s (1967) two Fox Hills members in the Fairpoint area, consists primarily of sandstone with large iron stained concretions and an upper unit of shales, siltstones, and sandstones, brightly colored by post-depositional paleosol development (Retallack, 1983; Jannett & Terry, 2008).

Figure 4 shows a stratigraphic column of the White Owl, SD, recovery site. Because our bone locality lies near the top of a hill (Fig. 2), beds significantly higher in the sequence than the bone horizon have been removed by erosion. Thus, the stratigraphic column for our recovery locality (Fig. 4) does not contain the upper part of the Fairpoint Member (Stoneville Lithofacies) or the White Owl Creek Member. The theropod site discussed here lies in the Fairpoint Member, about 40 m above the contact with the Pierre Shale (Fig. 4). Pettyjohn (1967) states that his enigmatic dinosaur bones were found in a channel deposit at the contact between what he considered the lower and middle parts of the Fairpoint Member. The approximate stratigraphic position of this bone bearing channel lies about 20 m below our theropod hoizon at White Owl. However, the actual site of Pettyjohn’s (1967) bone discovery is about 45 km northwest of our site.

Figure 4 Stratigraphic column of the Fox Hills Formation at the White Owl, SD, recovery site of DMNH EPV.138575.

Stratigraphic column of the Fox Hills Formation at the White Owl recovery site of DMNH EPV.138575 in the Fairpoint Area of western South Dakota. X: recovery horizon of DMNH EPV.138575. Scale bar = approximately 5 m. Note that the upper part of the Fairpoint Member of the Fox Hills Formation, and all of the White Owl Creek Member are missing due to erosion.

YPM VP.061705: Waage (1968) defines the Fox Hills Formation in the northcentral part of South Dakota (the “Type Area” in Corson, Dewey, and Ziebach counties) as consisting of the Trail City, Timber Lake, and Iron Lightning members, the latter of which Waage (1968) created by combining two sandy lithofacies characteristic of the upper part of the Fox Hills Formation in the Type Area. Speden (1970) and Landman & Waage (1993) used this tripartite stratigraphic framework as the basis of their investigations of the Fox Hills Type Area bivalve and ammonite faunas. The Trail City is the lowermost of these members and, according to Waage (1968), its thickness varies from about 21 m in the eastern part of the Type Area to about 70 m in the west. It consists primarily of fine clayey silt and contains richly fossiliferous concretionary horizons (Waage, 1968, Figs. 24, 25 and 26). The Trail City Member is distinguished from the Pierre Shale below it by its higher silt content and the presence of jarosite beds in many localities.

The Timber Lake Member consists primarily of sandstone locally variable in grain size, clay content and bedding. It too contains horizons preserving abundant fossil-rich concretions. The Timber Lake Member also varies in thickness across the Type Area. More than 30 m thick in central Dewey County, it rapidly pinches out westward and is no longer present in western Corson and Ziebach counties (Waage, 1968, Fig. 20). The contact of the Timber Lake Member with the Trail City Member below tends to be gradational, but southwestward in the Type Area the contact can often be recognized in terms of distinctive jarosite beds. Together with the Trail City Member, the Timber Lake Member represents a wedge-shaped sand body migrating southwestward into the shallow Western Interior Seaway near the close of the Cretaceous (Waage, 1968; Landman & Waage, 1993).

As conceived by Waage (1968), the Iron Lightning Member, the uppermost of the three Fox Hills members, consists of two contrasting sandy lithofacies, both of which differ from the sandy, clayey members of the Fox Hills Formation below it. The Bullhead Lithofacies consists primarily of finely bedded sand and silty clay usually having a brown color. The Colgate Lithofacies is a white to gray, lithic sandstone commonly occurring in lenticular bodies often showing prominent cross-bedding and large, dark colored, ovoid concretions. It also contains channel deposits, frequently with coarse debris, including fossils, preserved in the base of the channels. As described above, it is in one of these channel deposits about 14 m below the base of the overlying Hell Creek Formation, in which YPM VP.061705 was collected (Waage, 1968, pg. 133). Figure 5 shows a stratigraphic column of the beds exposed at the recovery site and the position of YPM VP.061705 in the section.

Figure 5 Stratigraphic column of the Fox Hills Formation at YPM VP.061705 recovery site.

Stratigraphic column of the Fox Hills Formation in Ziebach County at the type locality for the Iron Lightning Member. (A) Recovery horizon of YPM VP.061705. Lithology and thickness data from Waage’s measured section (Waage, 1968; pg. 132–134) and from Waage (1968, Fig. 10). Note that the intermediate Timber Lake Member is missing due to its westward pinching out mentioned in the text.

Waage’s (1968) stratigraphic sections from different parts of the Type Area (e.g., Waage, 1968; Figs. 10, 25, and 26; Landman & Waage, 1993; Fig. 3) show that sand bodies of the two lithofacies are interspersed irregularly throughout the Iron Lightning Member. Lithologically, the Iron Lightning Member resembles Pettyjohn’s (1967) Fairpoint Member in western south Dakota, and probably represents the later eastward migration of the Sheridan Delta near the close of the Maastrichtian rather than the westward advance of sedimentation of the Trail City and Timber Lake members.

Geologic age

General Considerations: The absence of distinctive, time-indicative fossils, ammonites in particular, in the Fox Hills Formation of the Fairpoint area of South Dakota has historically been a major impediment to building a solid understanding of Fox Hills age relationships in this area. It also obfuscates correlation of Fairpoint area lithology with that of the Type Area—a point recognized by both Waage (1968) and Pettyjohn (1967). Our aim here is to suggest probable time equivalencies in Fox Hills strata across South Dakota, particularly as they relate to the Fairpoint area. The key to this approach is to apply the ammonite range zones for the Western Interior to Fox Hills exposures in South Dakota, and to link these to radiometric age dates and magnetostratigraphic data where possible. This is done in Fig. 6, which is based on a large body of work identified in the caption to this figure. Figure 6 indicates that Fox Hills strata occur higher in the section and more recently in time toward the northeast, a situation that undoubtedly reflects the retreat of the Western Interior Seaway during the late Maastrichtian and Danian (Gill, Cobban & Kier, 1966).

Figure 6 Generalized biostratigraphy of the Fox Hills Formation across western South Dakota.

Generalized biostratigraphy of the Fox Hills Formation across western South Dakota, showing changes in positioning of the Fox Hills Formation, and its members relative to ammonite range zonation for the Western Interior. Black star—recovery horizon of the Fairpoint specimen (DMNH EPV.138575). White star—recovery horizon of the Yale specimen (YPM VP.061705). Stratigraphic section for Timber Lake is a composite for the Fox Hills Formation in the Type Area. Stratigraphic columns uncapped at the top by a line indicate that they are terminated at the top by erosion. Inset (A) shows the geographic locations of the four reference localities. Figure 6 is based on reconstructions from Landman et al. (2013, Fig. 5; Landman et al. 2019; Fig. 2); and Witts et al. (2022; Fig. 2); and on references therein.

Geologic Age of DMNH EPV.138575: Landman et al. (2013) point out that three of the exposures in Fig. 6 (Red Bird, Creighton, Timber Lake) contain index ammonite assemblages sufficient to establish the boundaries of the four ammonite zones given in Fig. 6 for these three sites. In contrast, the White Owl exposure, which preserves DMNH EPV.138575, like the Fairpoint area in general, contains no ammonites at all. Nevertheless, there is evidence suggesting that the White Owl horizon containing DMNH EPV.138575 lies in the middle to upper H. nicolletii Zone as diagrammed in Fig. 6. This interpretation derives in part from exposures at Creighton, SD, studied in detail by Chamberlain et al. (2001), Jannett & Terry (2008) and Landman et al. (2013). This work suggests strong parallels in lithology with Fox Hills beds in the Fairpoint area. Pettyjohn (1967) also noticed the same similarity in lithologic character between the Fairpoint area and Fox Hills strata exposed in the bluffs south of the Cheyenne River in and around Creighton. It should also be noted that the nearly flat-lying Fox Hills beds at White Owl lie at elevations, which if extended southward across the Cheyenne River would bring them into line with the Fox Hills units around Creighton. This equivalency would imply that the Fairpoint Member lies within the time interval represented by the H. nicolletii Zone as defined at Creighton by Landman et al. (2013), and as diagrammed in Fig. 6.

Another point is that Pettyjohn (1967) states that in the Fairpoint area the base of the Fox Hills Formation lies about 7 m above the Baculites clinolobatus Zone in the uppermost Pierre Shale. This is also the case at Creighton. In the Fox Hills Type Area, the base of the Fox Hills Formation is about 80 m above the B. clinolobatus Zone (Landman & Waage, 1993). These differences in relative positioning of the Fox Hills/Pierre contact, as noted in Landman et al. (2013, Fig. 5), suggest that the middle to upper Fairpoint Member in the Fairpoint area corresponds to the H. nicolletii Zone in the Fox Hills Formation Type Area.

We interpret these observations to mean that the Fairpoint horizon from which our specimen derives is from the lower part of the upper Maastrichtian sequence in western South Dakota. Linking this view to the radiometric age data of Cobban et al. (2006), Merewether, Cobban & Obradovich (2011), and Lynds & Slattery (2017), and following Witts et al. (2022) in assigning a dating framework of Western Interior ammonite zonation, we estimate that the approximate age of DMNH EPV.138575 is on the order of 69 Ma.

YPM VP.061705: The fact that the geologic age of Fox Hills beds decreases to the east in South Dakota means that the age of the Yale specimen Waage (1968) recovered in the Fox Hills Type Area is likely to be younger than the Fairpoint specimen even though both occur in sandy Colgate style lithologies.

YPM VP.061705, as reported by Waage (1968), was found at the base of a channel cut in the Iron Lightning Member about 14 m below its contact with the overlying Hell Creek Formation. In the Type Area, the H. nebrascensis Ammonite Zone, which overlies the H. nicolletii Zone, extends from just below the top of the Timber Lake Member, through the Iron Lightning Member, and into the overlying Hell Creek Formation (see Fig. 6) where remains of the signature species, H. nebrascensis, have been found in the Breien Member of the Hell Creek Formation (Hartman & Kirkland, 2002; Landman, in Hoganson & Murphy, 2002; N. Landman, 2022, personal communication). Since the Breien Member lies about 2 to 9 m above the contact with the Fox Hills Formation (Hoganson & Murphy, 2002), the top of the H. nebrascensis Zone is about 16 to 23 m above the Iron Lightning Member horizon containing YPM VP.061705. This places YPM VP.061705 squarely in the upper H. nebrascensis Zone, and thus makes it significantly younger than DMNH EPV.138575. How much younger is more difficult to determine due to geographically variable thicknesses and ages of the beds in question. However, as suggested in Fig. 6, magnetostratigraphy provides a clue. The magnetostratigraphy data of Hicks et al. (2002, Figs. 11, 13) from southwestern North Dakota and the data of Lund, Hartman & Bannerjee (2002; Fig. 10) from southcentral North Dakota, suggest that the base of the C30n polarity chron, which Witts et al.’s (2022) range data indicate to be about 68 Ma, lies about 10 m below the Hell Creek/Fox Hills contact. This is roughly the position of the bed containing YPM VP.061705 (14 m below the Hell Creek/Fox Hills contact). Thus, the age of this bone would probably be in the range of slightly more than 68 Ma, or nearly 1 Myr younger than that of DMNH EPV.138575.

SYSTEMATIC PALEONTOLOGY

Dinosauria Owen 1842 sensu Padian and May 1993

Theropoda Marsh, 1881 sensu Gauthier 1986

Tetanurae Gauthier, 1986 sensu Sereno et al. 2005

Coelurosauria von Huene, 1914 sensu Sereno et al. 2005

Description.—DMNH EPV.138575: The Fairpoint phalanx is relatively robust and proximodistally elongate, with a preserved length of 80 mm (see Fig. 7 and Table 1). Its proximal dorsoventral and mediolateral widths are slightly greater than its distal dorsoventral and mediolateral widths. The proximal articular facet is concave and subtriangular in proximal outline and overgrown with framboidal pseudomorphic (pyrite) hematitic concretions. These concretions conceal detailed morphological features over much of the proximoventral portion, and where mechanically removed dorsally, have invaded and dissolved much of the cortical surface and dorsal morphology. Similarly, the distal articular facet, including the distal-most section of the condyle, has been obliterated. We suspect that if the distal condyle were intact, the minimum length of the phalanx would likely be closer to 85 mm, and perhaps even greater.

Figure 7 Fairpoint Phalanx: DMNH EPV.138575.

Fairpoint phalanx, White Owl, South Dakota; DMNH EPV.138575. (A) Lateral, (B) dorsal, (C) distal, (D) medial, (E) ventral and (F) proximal views. Note that medial and lateral are provisional designations. DMNH EPV.138575 is tentatively assigned to Ornithomimidae. Scale bar is 5 cm. Photography by Katja Knoll.

Table 1 Mensuration data for theropod phalanges DMNH EPV.138575 and YPM VP.061705.

Character	DMNH EPV.138575	YPM VP.061705	
Length	Proximodistal	80	45	
Distal Width	Dorsoventral	24	19	
	Mediolateral	28	22	
Proximal Width	Dorsoventral	35	19	
	Mediolateral	27	24	
Midshaft Width	Dorsoventral	25	12	
	Mediolateral	27	18	
Shaft O		82		
Medial fossa	Depth	6		
Proximodistal Ø	15	7	
Dorsoventral Ø	11	5	
Lateral fossa	Depth	3		
Proximodistal Ø	10	6	
Dorsoventral Ø	6	4	
Note:

Measurements in mm. O is circumference Ø is diameter. Because YPM VP.061705 could not be examined in person due to pandemic restrictions in place during this research, some measurements could not be obtained.

The shaft is arched dorsoventrally and slightly mediolaterally constricted just proximal to the plane corresponding with the arch’s apex. In dorsal and ventral views, the shaft appears very slightly curved toward the larger collateral ligament fossa (here tentatively identified as the medial fossa), and neither proximal nor distal articular areas expand laterally. Just proximoventral to the larger (medial) ligament fossa, there is a slight protuberance that extents to the ventral surface of the bone. The shaft is oval in cross section, and in medial and lateral views, broadens toward the proximal end. Apart from longitudinal and transverse fractures and some spalling of the dorsal cortex, the shaft is in better condition than both distal and proximal ends. The ventral surface of the specimen is moderately flattened and slightly asymmetrical, and includes two parallel plantar ridges running longitudinally towards the proximal facet. The medial of the two ridges is confluent with a subtriangular, rugose plantar surface. Distally, the ventral surface is weakly indented with a circular post-condylar depression. The dorsal surface of the shaft features a relatively rounded, gently medially curved ridge that runs longitudinally, becoming more exaggerated toward the proximal end. A shallow depression of the extensor fossa is located dorsally, just behind the distal condylar surface.

The distal end is subrectangular in cross section, bearing a smooth, rounded articular condyle where not lost to diagenesis. The ventral margins of the distal condyle exhibit some pitting where true rugosities are apparent. The lateral collateral ligament pits are well developed, asymmetrical, teardrop shaped, relatively deep, and large, the medial pit being significantly larger than its counterpart. Although some erosion has occurred, there is no indication that the dorsal surface of the condyle is significantly narrower than the ventral surface, thus the collateral fossae are not clearly visible in dorsal view. In addition, there is a very slight oval depression just distal to the proximal articular facet on what we identify as the lateral side (the side featuring the smaller ligament fossa).

Comparisons. DMNH EPV.138575: The Fairpoint phalanx shares similarities with those reported from various tyrannosaurid taxa. These include a slightly sloping long axis; ventral rugosities near the proximal articular facet; and an arched ventral surface in medial and lateral views (Brochu, 2003); a shallowly concave proximal facet (evident in proximal phalanges; Lambe, 1917); deep and asymmetric collateral ligament fossae, a trait shared by all phalanges apart from those belonging to digit III, which are equal in size (Lambe, 1917; Brochu, 2003, Figs. 107 and 108; Brusatte, Carr & Norell, 2012); and a relatively shallow extensor pit (Brusatte, Carr & Norell, 2012, Fig. 80; Brochu, 2003, Fig. 105). However, there are also dissimilarities, some of which could be due to ontogenetic differences: DMNH EPV.138575 lacks the expanded distal and proximal articular regions relative to the shaft in dorsal and ventral views apparent in the adult Tyrannosaurus rex (Brochu, 2003; FMNH PR2081) but also, albeit less pronounced, in the subadult tyrannosaurid Albertosaurus sarcophagus from the Maastrichtian age Horseshoe Canyon Formation of Alberta, Canada (Mallon et al., 2019; CMN 11315); and the ratio of proximodistal length to mediolateral midshaft width is larger than three, if the abraded distal condyle is taken into account, a value greater than that identified for Tyrannosauridae (Brusatte et al., 2010, SOM).

Altogether, the Fairpoint phalanx appears more gracile than the pedal phalanges of adult tyrannosaurs known from North America of a similar stratigraphic age. Thus, we suspect that DMNH EPV.138575, if indeed tyrannosaurid in nature, could potentially have belonged to a subadult individual. In fact, DMNH EPV.138575 bears striking similarities with pedal phalanx II-2 of the subadult tyrannosaurid A. sarcophagus (CMN 11315) from the Horseshoe Canyon Formation in that both have a shallow extensor fossa, deep collateral ligament pits, and a minimally mediolaterally constricted and slightly curved shaft (Mallon et al., 2019, Fig. 16). However, CMN 11315 exhibits a proximally projecting dorsal lip at the proximal articular facet, a feature that is not apparent in DMNH EPV.138575 but may have been destroyed as the bone shows damage here. Also, as observed in the phalanges of many other tyrannosaurids, the distal condyle in CMN 11315 narrows dorsally, revealing the collateral fossae in dorsal view and resulting in a subtrapezoidal cross section rather than a subrectangular one as seen in our specimen.

Pedal phalanx II-1 of the subadult tyrannosaurid cf. Teratophoneus sp. (14UTKA-8-15-15F-15, Fig. 8; second column (G–L)) of the Campanian Kaiparowits Formation is comparable to the Fairpoint phalanx, as well. Both have a similar width to length ratio; deep collateral ligament fossae; a dorsoventrally arched shaft; and a shallow postcondylar depression. 14UTKA-8-15-15F-15, however, has significantly more pronounced plantar ridges; a deeper extensor fossa on the dorsal surface; expands slightly distally but especially proximally in relation to the shaft in dorsal view; its distal condyle is slightly narrower dorsally than ventrally; and its proximal facet is more circular and less triangular in outline than that of DMNH EPV.138575.

Figure 8 Fairpoint phalanx DMNH EPV.138575 compared to other theropod phalanges.

Fairpoint phalanx DMNH EPV.138575 compared to other theropod phalanges which have not been figured in existing literature. First column: DMNH EPV.138575, in (A) lateral, (B) medial, (C) dorsal, (D) ventral, (E) distal and (F) proximal views. Second Column: Pedal phalanx II-1 of the subadult cf. Teratophoneus sp. from the Campanian age Kaiparowits Formation, 14UTKA-8-15-15F-15, in (G) lateral, (H) medial, (I) dorsal, (J) ventral, (K) distal and (L) proximal views. Third Column: Pedal phalanx III-2 of a large-bodied ornithomimid from the Maastrichtian age Horseshoe Canyon Formation, TMP2015.007.0015, in (M) lateral, (N) medial, (O) dorsal, (P) ventral, (Q) distal and (R) proximal view. Scale bar is 5 cm. Photography by Katja Knoll (A–L); and by Rhian Russell (M–R), courtesy of the Royal Tyrrell Museum.

The Fairpoint phalanx also possesses morphological features observed in the proximal pedal phalanges of various ornithomimids. These characteristics, which are also represented in tyrannosaurids, include a shallowly concave proximal articular facet (Kobayashi & Barsbold, 2005; Cullen et al., 2013; Osmólska, Roniewicz & Barsbold, 1972; Chinzorig et al., 2017); a shallow extensor fossa (Shapiro et al., 2003, Fig. 1; Cullen et al., 2013, Figs. 2 and 3; Sues & Averianov, 2016, Fig. 24; Claessens & Loewen, 2015, Figs. 5, 6 and 8; McFeeters et al., 2016); and deep and distinct collateral ligament fossae (Smith & Galton, 1990; Kobayashi & Barsbold, 2005; Shapiro et al., 2003, Fig. 1).

Although DMNH EPV.138575 is significantly larger, the specimen shares similarities with phalanx II-1 of Ornithomimus velox of the late Maastrichtian Denver Formation (Claessens & Loewen, 2015; Figs. 5, 6 and 8; YPM 548). These include a slightly deflected ridge running longitudinally across the dorsal surface; deep, asymmetrical collateral ligament pits; a slight protuberance emanating just proximoventrally from the larger ligament pit; and a shallow extensor fossa. However, in DMNH EPV.138575 the proximal articular facet is more triangular in outline when viewed proximally, and the ventral ridges near the proximal facet are notably less pronounced, though possibly abraded.

Morphologically very similar and more size-equivalent with the Fairpoint phalanx are pedal phalanges III-1 and III-2 belonging to a large-bodied ornithomimid (CMN 12068) from Canada’s Maastrichtian section of the Horseshoe Canyon Formation and pedal phalanges III-1 and III-2 of Struthiomimus (TMP 1998.026.1) from the late Maastrichtian Scollard Formation. While the proximal end is laterally slightly more expansive relative to shaft width in III-1 and III-2 of both CMN 12068 and TMP 1998.026.1, all are straight shafted, relatively robust and elongate, have a similar width to length ratio, shallow dorsal extensor ligament fossae, and the dorsal surface of the distal condyle does not notably narrow (Cullen et al., 2013, Fig. 2; Nottrodt, 2021, Fig. 7). The Fairpoint phalanx also compares closely with right pedal phalanx III-2 of an ornithomimid foot from the Maastrichtian age Horseshoe Canyon Formation of Alberta, Canada, (TMP2015.007.0015; see Fig. 8; third column (M–R)) and pedal phalanx III-1 of a large-bodied ornithomimosaur from Mississippi’s Santonian age Eutaw Formation (Chinzorig et al., 2022, Fig. 7C, MMNS VP-4949). These characteristics include an overall cylindrical, largely symmetrical and elongate shape; a shallow extensor ligament pit; and steep sided lateral surfaces of the proximal region. Both TMP2015.007.0015 and DMNH EPV.138575 also feature an elongated lateral tuberosity adjacent to a broad subtriangular medial tuberosity on the ventral surface near the proximal facet, and a shallow postcondylar depression on the ventral surface. Conversely, the expansion of both distal and proximal regions relative to the shaft in dorsal view set TMP2015.007.0015 and the Eutaw Formation specimen (MMNS VP-4949) apart from DMNH EPV.138575. Other anomalously large ornithomimid elements are known from other Cretaceous deposits in North America, including the Dinosaur Park Formation (Longrich, 2008), suggesting the presence of unidentified large-bodied taxa or upper body size limits beyond expectations based on more complete materials.

Because no other skeletal elements have been found associated with DMNH EPV.138575, and because some key morphological features are either destroyed or concealed by hematitic overgrowths, we cannot conclusively assign the element to a particular non-avian theropod clade. However, based on several morphological characteristics, size of the element and stratigraphic age, we tentatively attribute the phalanx to a member of Coelurosauria, likely pedal phalanx III-2 belonging to a large-bodied ornithomimid.

Dinosauria Owen 1842 sensu Padian and May 1993

Theropoda Marsh, 1881 sensu Gauthier 1986

Tetanurae Gauthier, 1986 sensu Sereno et al. 2005

Coelurosauria von Huene, 1914 sensu Sereno et al. 2005

Description.—YPM VP.061705: While relatively robust, phalanx YPM VP.061705 is significantly smaller than DMNH EPV.138575, with a proximodistal length of 44 mm (see Fig. 9 and Table 1). The phalanx preserves much of its original surface, missing only the dorsal half of the proximal articular surface to breakage. A weak, vertical medial ridge divides the proximal articular cotyle into slightly concave, subequal medial and lateral portions. In proximal view, the articular surface appears sub-triangular to moderately pentagonal in cross-section, with the lateral sides nearly vertical (steep sided). Ventrally on the proximal end, a broad lip-like asymmetrical flange projects medially. The proximoventral surface is planar, with two faint plantar ridges oriented longitudinally near the proximal facet. The cortical surface is highly vascularized here. The shaft is moderately arched dorsoventrally and mildly pinched near the distal condyle in lateral view. The proximal and distal areas are expanded relative to the shaft in dorsoventral but only slightly so in mediolateral view. The distal articular condyle is divided into two highly asymmetrical hemi-condyles separated by a vertical sulcus. The medial hemi-condyle is dorsoventrally significantly larger than its lateral counterpart, dorsally thickened, and inclined dorsolaterally towards the sagittal midline. The lateral and medial ligament fossae are ellipsoidal to subcircular, the medial being deeper and modestly visible in dorsal view. On the dorsal surface just proximal between both hemi-condyles, a relatively shallow extensor fossa is evident.

Figure 9 Iron Lightning phalanx, YPM VP.061705.

Iron Lightning phalanx, YPM VP.061705, Ziebach County, South Dakota. (A) Lateral, (B) dorsal, (C) distal, (D) medial, (E) ventral and (F) proximal views. Note that medial and lateral are tentative designations. YPM VP.061705 is tentatively assigned to Ornithomimidae. Scale bar is 5 cm. Courtesy of the Division of Vertebrate Paleontology; Peabody Museum of Natural History, Yale University; Photography by Vanessa R. Rhue (CC-0).

Macroscopically, YPM VP.061705 appears well-preserved. As in other vertebrate appendicular elements capped by cartilaginous soft tissues, the texture of the articular surfaces is distinctly roughened and contrasts with the smooth, compact cortical surface of the shaft. Some mild pitting of the cortex is apparent but whether this is diagenetic or pathologic is unclear. Where the bone is spalled, the internal spongy tissue is porous, showing little to no diagenetic mineral infilling.

Comparisons. YPM VP.061705: Because many morphological traits of pedal phalanges are shared by various coelurosaurian clades, particularly between the temporally relevant members of Ornithomimidae and Tyrannosauridae, it is difficult to attribute this single, isolated specimen to either group definitively. Nevertheless, because of its geographic and stratigraphic location, its general morphology and size, we cautiously assign YPM VP.061705, like DMNH EPV.138575, to Ornithomimidae, and believe it may represent pedal phalanx II-2. Notwithstanding, similarities with tyrannosaurid pes material are undeniable. For example, YPM VP.061705 closely resembles pedal phalanx II-2 of the subadult tyrannosaurid cf. Teratophoneus sp. (14UTKA-8-15-30, Figs. 10G–10L). Both are similarly proportioned, and like YPM VP.061705, 14UTKA-8-15-30 has a slightly dorsoventrally arched shaft; a sub-triangular proximal facet in proximal view, divided by a gentle keel into two asymmetrical concavities; a proximoventral flange on the medial side; and two faint plantar ridges near the proximal facet. Differences between both specimens are mostly confined to the distal region: in 14UTKA-8-15-30, the collateral ligament fossae are located more dorsolaterally within the distal hemi-condyles and are more teardrop shaped with the long axes paralleling the long axis of the phalanx, whereas the ligament pits in YPM VP.06170 are placed more centrally within their respective hemi-condyles and are somewhat more circular in outline. Lastly, the dorsal surface of the distal condyles is notably narrower relative to the ventral surface in 14UTKA-8-15-30 compared to that of YPM VP.06170; and in distal view, the two hemi-condyles are markedly asymmetrical in YPM VP.06170 with the medial one being significantly larger than the lateral one, whereas this asymmetry is only gently expressed in 14UTKA-8-15-30.

Figure 10 Iron Lightning phalanx, YPM VP.061705, compared to other theropod phalanges.

Which have not been figured in existing literature. First Column: Iron Lightning phalanx, Ziebach County, South Dakota, YPM VP.061705, in (A) lateral, (B) medial, (C) dorsal, (D) ventral, (E) distal and (F) proximal views. Second Column: Pedal phalanx II-2 of the subadult cf. Teratophoneus sp. from the Campanian age Kaiparowits Formation, 14UTKA-8-15-30, in (G) lateral, (H) medial, (I) dorsal, (J) ventral, (K) distal and (L) proximal views. Scale bar is 5 cm. Photography Courtesy of the Division of Vertebrate Paleontology; Peabody Museum of Natural History, Yale University; Vanessa R. Rhue (A–F) (CC-0); and by Katja Knoll (G–L).

Several attributes are consistent with pedal phalanx II-2 of two large-bodied ornithomimids: CMN 12068 from Canada’s Maastrichtian-age section of the Horseshoe Canyon Formation (Cullen et al., 2013); and TMP 1998.026.1 belonging to a Struthiomimus from the late Maastrichtian Scollard Formation (Nottrodt, 2021). These include width to length ratio, a pronounced proximoventral flange, and the slight expansion of both proximal and distal ends in dorsal view (Cullen et al., 2013, Fig. 2; Nottrodt, 2021; Fig. 7). In addition, both YPM VP.06170 and TMP 1998.026.1 have ellipsoidal collateral ligament fossae, a ginglymoid proximal articular facet, and, while the proximal facet in proximal view is more rounded dorsally (D shaped) in TMP 1998.026.1, both specimens feature lateral and medial sides that are nearly vertical (Nottrodt, 2021; Fig. 7). Likewise, we see considerable similarities with pedal phalanx II-2 of a large lower Cretaceous ornithomimid from China’s Gansu province (IVPP V12756) and pedal phalanx II-2 belonging to the ornithomimid Aepyornithomimus tugrikinensis of the Late Cretaceous Djadokhta Formation of Mongolia (MPC-D 100/130). These include deep collateral ligament fossae placed centrally within their respective, significantly enlarged condyles in medial and lateral views; a shallow extensor fossa on the dorsal surface; and overall shape and width to length ratio (Shapiro et al., 2003; Figs. 1C and 1D; Chinzorig et al., 2017, Fig. 4).

Pedal phalanx II-2 of the ornithomimid Rativates evadens from the upper Campanian Dinosaur Park Formation (ROM 1790) and pedal phalanx IV-2 (MMNS VP-7119) from a large-bodied ornithomimosaur of the Santonian age Eutaw Formation also resemble YPM VP.06170. Like YPM VP.06170, ROM 1790 and MMNS VP-7119 exhibit a broad lip-like, medially projecting asymmetrical flange on the ventral surface of the proximal end; an asymmetrical distal condyle in distal view; have a keel dividing the proximal articular facet; and centrally placed collateral ligament fossae (McFeeters et al., 2016, Fig. 11; Chinzorig et al., 2022, Fig. 7D). ROM 1790, however, is significantly shorter and stouter than YPM VP.06170; the constriction of its shaft is less pronounced just proximal of the distal condyle; its ligament fossae are more enlarged; and it is laterally more compressed, which results in a narrow proximal articular facet in proximal view (McFeeters et al., 2016, Fig. 11). The Eutaw Formation specimen distinguishes itself from YPM VP.06170 by featuring a more pronounced marginal ridge on the dorsal surface of each distal hemi-condyle; enlarged lateral ligament pits; and an expanded distal condyle relative to mid-shaft diameter (Chinzorig et al., 2022, Fig. 7D).

Dissimilarities in pedal phalanx II-2 of the stratigraphically similar North American Ornithomimus velox (YPM 548) indicate either a tentative assignment of YPM VP.061705 to Ornithomimidae or reveal undocumented variation in pedal phalangeal morphology within the clade. Phalanx II-2 of O. velox is stouter with a greater width to length ratio than YPM VP.061705, and a notably shorter shaft, a less exaggerated distal condyle relative to shaft diameter in lateral view, a shallower ligament fossa, and a more acute narrowing of the distal condyle dorsally (Claessens & Loewen, 2015; Fig. 8).

Discussion

Depositional Environment of DMNH EPV.138575: Landman et al. (2013; Fig. 5) indicate that the western shoreline of the Western Interior Seaway (WIS) during the upper H. nicolletii Zone extended from SW to NE across South Dakota. Our collection site at White Owl, South Dakota, lay very close to, but on the seaway side, of the shoreline. They envision this shoreline as highly irregular and characterized by headlands, bays, estuaries, bars, and shoals, a view consistent with Hoganson, Erickson & Holland’s (2007) description of the shoreline in North Dakota, and that of Becker, Chamberlain & Terry (2004) for the paleoenvironment of the collection site itself.

The preservation of fossils, including DMNH EPV.138575, at the White Owl locality in sandstones showing prominent cross-bedding throughout the exposure (Figs. 3A, 3B; and Fig. 4) suggest that the fossils found here probably accumulated in a nearshore marine setting with a water depth near wave base. While deposition in a shifting, unstable, sandy nearshore environment seems apparent, Becker, Chamberlain & Terry (2004) also note that except for the beds shown in Fig. 2, from which their specimens, and DMNH EPV.138575, derive, fossils are uncommon above and below the bone horizon. This, together with the observation that this assemblage consists of mixed terrestrial, freshwater, and marine faunal elements, suggests that the fossiliferous beds at White Owl may represent a condensed section resulting from a short-lived transgressive event in an otherwise overall sea-level regression associated with the retreat of WIS waters at the close of the Cretaceous and the development of the Dakota Isthmus (Erickson, 1978, 1999). Similar mixed assemblages deriving in part from short term transgressive events, are known from the Atlantic Coastal Plain, such as in the Campanian Black Creek Group of North Carolina (Schwimmer, 1997) and the Campanian Marshalltown Formation at Ellisdale, New Jersey (Brownstein, 2018). Such occurrences are known also from the WIS, as in the case of the Campanian Dinosaur Park Formation of Alberta (Eberth, 1996; Eberth & Brinkman, 1997). The discovery of scaphite shell fragments in the Hell Creek Formation (Hoganson & Murphy, 2002; Hartman & Kirkland, 2002) and Lance Formation (Jeletzky & Clemens, 1965), and mixed freshwater-marine fossil assemblages from the Type Area Iron Lightning member of the Fox Hills Formation itself (Waage, 1968), indicate that transgressive marine incursions are a feature of the waning phases of the WIS. Deposition of the bed at our White Owl recovery locality preserving the Fairpoint phalanx was probably produced by a more transient, small-scale transgression than those observed in the continental Hell Creek or Lance Formations, and one focused more offshore than these other WIS and Atlantic events.

Depositional Environment of YPM VP.061705: The depositional environment preserving the Yale phalanx is also marginal marine, but it is not the same as the shallow, nearshore paleoenvironment in which the Fairpoint phalanx occurs. Waage (1968) regarded the Iron Lightning Member as the product of coastal, lagoonal, and delta-topset deposits related to the eastward migration of the Sheridan Delta. Channels of Colgate lithology were cut into these deposits by currents flowing across them. Waage (1968) points out that the Yale phalanx was part of a basal channel accumulation containing terrestrial, freshwater, and nearshore marine fossils. Thus, there is the possibility that YPM VP.061705 was preserved in a tidal or distributary channel associated with the deltaic setting then becoming widespread in the South Dakota region of the Western Interior. The depositional environment of YPM VP.061705 would appear, therefore, to have been somewhat more onshore as compared to that of DMNH EPV.138575.

Temporal Significance: The inferred age of 69 Ma for the horizons preserving DMNH EPV.138575 place it in a poorly-represented biochronological interval of the middle Maastrichtian, within the poorly defined ‘Edmontonian’ NALMA. Contemporary Western Interior terrestrial faunas from this interval are known from the Prince Creek Formation of Alaska (Mull, Houseknecht & Bird, 2003); the Wapiti (Unit 5; Fanti & Catuneanu, 2009) and Horseshoe Canyon (Tolman Member; Eberth & Braman, 2012) formations of Alberta; the North Horn Formation (Unit 1; Difley & Ekdale, 2002) of Utah; the Ojo Alamo Formation of New Mexico (Lucas et al., 2009), the Javelina Formation of Texas (Lehman, Mcdowell & Connelly, 2006); and possibly portions of the lower Laramie Formation of Colorado (Raynolds, 2002; Wilson, Dechesne & Anderson, 2010). Many of these faunas preserve similar dinosaurian components, including hadrosaurid, ceratopsid, pachycephalosaurian, and ankylosaurian ornithischians, and tyrannosaurid, ornithomimid, oviraptorosaurian, and paravian theropods. The presence of an ornithomimid, or tyrannosaurid, is therefore not surprising, though it does underscore the potential significance of any terrestrial vertebrate remains from the Fox Hills Formation in understanding biotic distribution and diversity of the Western Interior during the Edmontonian NALMA.

Taphonomy of DMNH EPV. 138575: The Fairpoint specimen appears to have undergone a complex taphonomic history which we suggest has two main phases: (1) temporary burial near the animal’s death site in an onshore, coastal environment; and (2) subsequent transport to its nearshore, marine recovery site, and final preservation in the sand-dominated depositional environment existing at the recovery site. Evidence supporting this interpretation lies in the physical attributes of the specimen and in the concretionary overgrowths that adhere to it.

1. Initial Burial: There are several longitudinal and transverse cracks in the cortical bone. These are visible in Fig. 7. Although some breakage occurs in acute or obtuse angles, the primary breakage pattern here appears to be orthogonal, at right angles. Such a pattern has been observed in the fracturing of dry bone, i.e., breakage that occurs in purely mineralized or permineralized bone after the loss of internal organic material (Johnson, 1985; Morlan, 1984; Villa & Mahieu, 1991). Originally, the specimen may thus not have been deposited directly in the marine setting where it was found because drying out would not have been possible in such an environment. Instead, the bone probably spent considerable time post-mortally in a dryland, coastal or estuarine locale.

2. Transport and Final Burial: The proximal and distal ends of the phalanx are damaged and partly hidden by concretionary overgrowths evident in Fig. 7. Nevertheless, where damage is minimal and the bone surface is visible, the bone does not appear to be abraded to any significant degree, even though it was buried in what was clearly an unstable, shifting substrate. This suggests that transport to the final nearshore resting place of the specimen was minimal, i.e., its initial deposition and final preservation site were nearby, and that exposure to current or waves during transport and burial was probably short-lived. Mineralization occurring early in the specimen’s diagenetic history may have been a factor here by hardening the bone material and enhancing its resistance to abrasion.

3. Was the Fairpoint Phalanx Carnivore-Consumed?: In a few places, slivers and flakes of cortical bone have spalled off the specimen. These features are also clearly seen in Fig. 7. This does not appear to be the result of predation or scavenging prior to desiccation and permineralization because the bone lacks the V-shaped, or U-shaped punctures and arcuate scrape marks often made by the teeth of some carnivorous dinosaurs when feeding (Erickson & Olson, 1996; Fowler & Sullivan, 2006; Hone et al., 2010; Hone & Tanke, 2015; Brown, Tanke & Hone, 2021). However, it is possible that the Fairpoint specimen was consumed without being bitten because, being small, the bone could easily have been swallowed whole, without contacting the teeth of the consumer, as might occur if the bone were well inside a chunk of flesh sliced off the carcass by a large carnivore. In addition, there is some evidence that many carnivorous dinosaurs avoided contacting bone with their teeth when feeding (Fiorillo, 1991; Hone & Rauhut, 2010). However, the Fairpoint bone also lacks the eroded, broken-down surface resulting from passage through the gut of a carnivore (Chin et al., 1998; Varricchio, 2001). It appears to us, therefore, that the bone was not carnivore consumed while on land. Aquatic carnivores make bite marks similar to those of terrestrial counterparts (Schwimmer, Stewart & Williams, 1997; Everhart & Ewell, 2006; Becker, Chamberlain & Goldstein, 2006) and similarly degrade bone during digestion (Everhart, 2003, 2004; Everhart & Ewell, 2006; Schwimmer, Weems & Sanders, 2015), so that the specimen is probably not the product of bloat and float nekroplanktonism. We think that the Fairpoint bone was long devoid of flesh when it reached an aquatic setting. It is thus unlikely that the Fairpoint phalanx was the target of carnivores on land or in the sea. We infer, therefore, that the flaking of cortical bone visible in Fig. 5 is related to desiccation of the bone occurring early in its diagenetic history.

Finally, we note that it is possible such bone flaking may have been due to the impact of large pebbles or other objects mobilized by storm or tidal flows in the specimen’s nearshore preservation locale, but the near absence of abrasion on the bone and the absence of large clasts in the sandstone argue against this alternative.

4. Concretionary Overgrowths: Small hematitic concretions adhere to the surface of the bone (Fig. 7). Two particularly evident hemispherical concretions, each about 1.5 cm in diameter, attach to the articulation surface at the proximal end of the bone (Fig. 7). The bone has a dark color where similar concretions have broken away from the bone surface on both proximal and distal extremities. In addition, smaller, flattened irregular concretions coat portions of the bone shaft and the surfaces of the collateral ligament pits. All of these concretions are composed of sedimentary grains, mostly quartz, cemented together and to the bone by microcrystalline hematite and probably other iron oxides as well. The sand grains within these concretions have the same size, shape, and sorting as the sand comprising the outcrop as a whole. These observations suggest that the concretions probably formed late in diagenesis as the bone lay in the sand where it was found and as part of the process that produced the large concretions and concretionary horizons seen in Fig. 2.

Yet, an argument can be raised suggesting that concretion formation may have been a two stage process in which iron-bearing concretionary growth began very soon after the animal’s death and prior to desiccation and transport. Pyrite and iron oxides can replace organic material (Sawlowicz & Kaye, 2006; Canfield & Raiswell, 1991), and can form in and on fossil bone in various different ways (Pfretzschner, 2001a, 2001b; Bao, Koch & Hepple, 1998). Decomposition of organic matter can nucleate concretions and spur their growth because of its effect on local pH and eH. The occurrence of the concretions associated with articulation surfaces and ligament pits is interesting because it is these parts of the bone to which tendon and cartilage, which are soft tissues slow to decay, are attached. The concretions visible on DMNH EPV.138575 thus may mean that flesh still adhered to it when it was initially buried. Alternatively, these concretions could have precipitated on those surfaces favorable to the decomposition of the bone’s less durable internal organic compounds. As is the case in most long bones (Bishop et al., 2018; Moreira et al., 2019), the cortex of pedal phalanges appears thickest along the shaft (where the mineral density is higher), whereas near the proximal and distal articulation surfaces the cortex progressively thins and the internal space is dominated by more vascular cancellous bone tissue whose mineral density is lower. Near or at the articulation surfaces these conditions (thin cortex and porous bone texture) would allow for easier access to bone internal organic material, such as collagen, enabling microbial decomposition, the byproducts of which (i.e., sulfide), if combined with dissolved iron, could have precipitated pyrite in and on the bone, and resulted in thin, framboidal pyritic coatings on the susceptible parts of the bone structure. This form of pyritization would have occurred during early diagenesis (Pfretzschner, 2001a, 2001b, 2004) before the bone dried out because the dry bone fracture pattern suggests the absence of organic material when breakage occurred. In our taphonomic model, this first stage of concretion formation with its thin pyritic coatings adhering to susceptible parts of the bone would thus have ended prior to desiccation, and before transport of the bone to the nearshore setting where it was buried and subsequently found.

The second stage of bone concretionary growth is unrelated to the first stage, and is synonymous with the production of the large hematitic concretions and concretionary horizons that dominate the bedding at the discovery site (Fig. 2). Concentrations of organic debris, sometimes with recognizable constituents (as in Fig. 3D), that occur within some of these large concretions point to the importance of biodegradation in their formation. However, DMNH EPV.138575 lacked organic material at this stage as we hypothesize above. Instead, we propose that it was residual iron crystallization from the initial, pre-desiccation, mineralization process in the bone that controlled the nucleation of second stage concretionary growths on the bone surface. This produced bone overgrowths physically similar to other concretions present at the recovery site but with a pattern of attachment to the specimen that is consistent with the diagenetic susceptibility of the fresh bone to early post-mortem pyritization.

Taphonomy of YPM VP.061705: We do not attempt to interpret the taphonomy of the Yale bone in detail because we were unable to examine the specimen first-hand due to COVID-19 pandemic restrictions in force at the time of the writing of this article. However, the overall exceptional preservation of the external bone surface, preserving minute details of vascularization and soft tissue attachments, suggest rapid burial with minimal transport.

CONCLUSIONS

In this article we report the discovery of two theropod pedal phalanges from marginal marine beds exposed at two widely separate localities of the Maastrichtian Fox Hills Formation in western South Dakota. The morphology of these bones suggests that they derive from large-bodied ornithomimids, although it is not possible to rule out juvenile tyrannosaurids as alternatives. The first of these bones was found in White Owl, Meade County, SD, in a heavily cross-bedded sandstone of the Fairpoint Member containing the remains of both aquatic and terrestrial organisms. The recovery site represents a sandy, near-shore depositional environment seaward of the shoreline and above wave base. The second bone, originally found by Waage (1968), but undescribed until now, is from an exposure of the upper Iron Lightning Member near the hamlet of Iron Lightning in Ziebach County, SD. The specimen comes from the base of a channel deposit, possibly a tidal channel, and was part of an assemblage of fragmentary marine and terrestrial organisms, including a variety of dinosaurs. Both specimens point to the idea that Laramidian ornithomimids (or tyrannosaurids) inhabited coastal Maastrichtian environments when alive. Because the specimens are from the marine Fox Hills succession, it is possible to establish their geologic ages using ammonite range zonation data. From this it is apparent that the White Owl specimen derives from the Hoploscaphites nicolletii Zone and thus has an approximate age of 69 Ma, while the Iron Lightning specimen is from the overlying H. nebrascensis Zone with an age of about 68 Ma. Both specimens therefore come from a time when the fossil record of Laramidian ornithomimids is relatively impoverished. In this, the Fox Hills discovery mirrors similar low abundances of contemporary ornithomimids on the Appalachian side of the Western Interior Seaway (e.g., Brownstein, 2018; Chinzorig et al., 2022).

Supplemental Information

Supplemental Information 1 Phalanx specimens.

Click here for additional data file.

We thank Alan Titus, Paria River District Paleontologist (Bureau of Land Management), and Neil Landman (American Museum of Natural History) for their comments on an earlier version of this article. We also appreciate the help of Vanessa Rhue (Yale Peabody Museum) in tracking down and providing photographs of YPM VP.061705, and of Rhian Russel (Royal Tyrrell Museum) for providing photographs of TMP2015.007.0015. We thank Sally Shelton and Jim Fox (Museum of Geology, South Dakota School of Mines), and Tim Cowman (South Dakota Geological Survey) for their assistance in our attempt to find Pettyjohn’s (1967) missing dinosaur material. We greatly appreciate the input of Denver Fowler (Badlands Dinosaur Museum), and a second, anonymous reviewer for this journal who helped us crystallize some of our ideas and improve their presentation. JAC also wishes to acknowledge the assistance of his field partners, particularly Marty Becker, Ashleigh Chamberlain, Matthew Garb, Patricia Jannett, Paula Messina, Phil Stoffer, and Dennis Terry, during the time in which the Fairpoint phalanx was collected.

Institutional Abbreviations and Repositories

CMN Canadian Museum of Nature, Ottawa, Ontario, Canada

DMNH Denver Museum of Nature and Science, Denver, Colorado, USA

FMNH Field Museum of Natural History, Chicago, Illinois, USA

IVPP Institute of Vertebrate Paleontology and Paleoanthropology, Beijing, China

MPC Mongolian Paleontological Center, Mongolian Academy of Sciences, Ulaanbaatar, Mongolia

PRD Paleontology Laboratory, Bureau of Land Management, Paria River District, Kanab, Utah, USA

ROM Royal Ontario Museum, Toronto, Ontario, Canada

TMP Royal Tyrrell Museum of Palaeontology, Drumheller, Alberta, Canada

UMNH Natural History Museum of Utah, Salt Lake City, Utah, USA

YPM Yale Peabody Museum, New Haven, Connecticut, USA

Additional Information and Declarations

Competing Interests

Author Contributions

Field Study Permissions

Data Availability

The authors declare that they have no competing interests.

John A. Chamberlain, Jr conceived and designed the experiments, performed the experiments, analyzed the data, prepared figures and/or tables, authored or reviewed drafts of the article, and approved the final draft.

Katja Knoll conceived and designed the experiments, performed the experiments, analyzed the data, prepared figures and/or tables, authored or reviewed drafts of the article, and approved the final draft.

Joseph J. W. Sertich conceived and designed the experiments, performed the experiments, analyzed the data, authored or reviewed drafts of the article, and approved the final draft.

The following information was supplied relating to field study approvals (i.e., approving body and any reference numbers):

Access to the property from which DMNH EPV.138575 was collected was provided by verbal agreement from the Hall, Mickelsen, and Petersen families who owned the land when the field work was done (1999–2002).

The following information was supplied regarding data availability:

The theropod phalanx from the Fairpoint Member of the Fox Hills Formation in White Owl, Meade County, South Dakota is deposited in the vertebrate paleontology collections of the Denver Museum of Nature and Science (DMNH), Denver, Colorado, USA, and is identified by the catalogue number: DMNH EPV.138575.

The theropod phalanx from the Iron Lightning Member of the Fox Hills Formation near Iron Lightning, Ziebach County, South Dakota is in the vertebrate paleontology collections of the Yale Peabody Museum (YPM), Yale University, New Haven, Connecticut, USA, and carries the catalogue number: YPM VP.061705.

Physical attributes of these specimens are described in Table 1, and images of them can be found in Figs. 7 and 9.

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
