# Peer review of "Non-avian theropod phalanges from the marine Fox Hills Formation (Maastrichtian), western South Dakota, USA"

_PeerJ, doi:10.7717/peerj.14665_

## Round 0.1 · original submission · Minor Revisions

The two reviewers have provided abundant, extremely constructive comments on the manuscript, and both find it to be a valuable and rigorous study; yet there are some critiques of the interpretation and a need for more comparisons/revisions of the text and figures. These all seem very achievable.

·

Basic reporting

WRITING AND STRUCTURE
The paper is written in clear English. I only have some minor issues. In the attached PDF, you’ll see in places I have deleted extraneous words. Throughout the manuscript there are often extraneous words, sentences, and possibly whole sections. I appreciate thoroughness, but sometimes I think you could just delete some of this. Partly, it derails my concentration when reading – overlong sentences with extraneous adjectives can make it difficult to follow the purpose of the section. Other times there just isn’t the need for the detail. I don’t think, for example, that we need much of a review on marine dinosaur remains. The entire review section could be drilled down to a couple of sentences

FIGURES
It’s a more general point, but could the scalebars on the figured fossils be more intuitive? I have never really understood the aversion to having “5 cm” written under a scale bar in a small font – it is distracting and dissociating to look over at the caption to see what the size is. An alternative is to provide the scale bar as 5 alternating black and white stripes thereby showing intuitively that it is 5mm, 5cm, or 5m (with cm being most likely).

The color photos in Figure 2 and 3 are very welcome. I note that most previous papers on these subunits have only black and white images. I find b/w images much less useful for recognizing stratigraphic units, so thank you for including these.

Figure 6: “Iron Lightening” is misspelled. Please fix and check document for further misspellings of this term.

Please could you add tentative identifications to the captions for figs 5 & 6.

FURTHER NOTES
I have some specific notes and questions added as comments on the PDF (see attached).

Experimental design

WHAT IS THE FOX HILLS
This is not a criticism of this paper. The authors put a lot of effort into trying to figure out the stratigraphic age of these specimens, and this therefore requires review of the Fox Hills, especially the different members in different states, and what these might correlate with.

I very much appreciate that this a sometimes maddeningly confusing subject, since we are basically looking at a prograding delta system that sometimes broadly moves West-East, other times other directions, also depending on where you are. There are issues in lithostratigraphically correlating facies which may or may not be chronostratigraphically comparable, never mind the awkwardness that what may in one state be called Fox Hills, is in another state considered part of the Hell Creek Formation, or Pierre Shale.

I have spent some time attempting to tease apart this issue (using sequence stratigraphy) at the bottom of the Montana Hell Creek in the type area (Fowler, 2020) and correlating across the region (Fowler, 2017), so I read with great interest the discussion on the e.g. Fairpoint and White owl Creek, and the Iron Lightning Members.

I’ve been looking through Becker (2004), Waage (1968), Pettyjohn (1967) and others, while reading through the current manuscript’s discussion on correlation between the type Fox Hills and these fossil localities, and one thing that I think is definitely needed is a set of geologic columns that show the inferred correlations and inferred or demonstrated stratigraphic ages. It is quite confusing to read these accounts of e.g. the Enning facies and where Black (1964) thinks they may correlate then cross reference this with Pettyjohn (1967)– I am fairly cognizant of these unit names and relative ages and I am getting confused reading it all as a block of text. People unfamiliar with these units will just get lost – they’re not going to remember the relative order of the Iron Lightning and Timber lake members, never mind internal units within the Iron Lightning (Colgate and Bullhead), and even more that in some other places the Colgate is broken out as a member in its own right, or even a Formation in Canada (where it is called the Whitemud).

This mass of terms and dates would be best illustrated in a figure with 3-8 columns with all the beds/members labeled, with any ash dates, magstrat, and biostrat (etc) plotted. I think that this is my only major criticism of the paper– and it’s just a case of adding this figure, then maybe some of the convoluted explanatory text can be illustrated instead visually and the text deleted, or at least simplified.

SPECIMENS
The descriptions of the specimens are fine as is; if anything more detailed than necessary. The identifications look about correct… although I think the YPM phalanx might be tyrannosaur (it could be ornithomimosaur too – I’m not 100% on either), it does not matter really for the sake of the paper.

Validity of the findings

The specimens are appropriate to publish as records of dinosaur material in the Fox Hills, which is unusual. The paper states that these are the first dinosaur bones from the Fox Hills (I think possibly just of SD, but I am not sure). This is possibly true – I know of unofficial reports of Fox Hills dinosaur bones in a few collections from Montana. I found the paper to be very interesting regarding the discussion of the “Fox Hills”, and what this might mean for correlation.

Additional comments

This manuscript describes two isolated theropod phalanges recovered from the Fox Hills Formation of South Dakota. The paper discusses the taphonomy, potential age and stratigraphic correlations of the specimens and localities, and gives various overviews on geography, marine preservation of dinosaurs, time significance etc.

I have annotated and edited the PDF in a few places. I have also asked a few questions posted as comments within the PDF – these mainly request clarification or rephrasing. This PDF is attached to this review.

Overall the specimens are worth describing as points of note for the Fox Hills Formation. I would like to see the writing tightened up a little, but my main request is for a figure showing the different stratigraphic columns and their relative ages and correlations (etc). Ultimately the significance of the specimens is based on their stratigraphy, so I think for this reason it needs to be clearer.

I wish to reveal my identity: Denver Fowler.

Reviewer 2 ·

Basic reporting

I have reviewed the article and found it very interesting. Description of dinosaur material, no matter how small, from marine formation that preserves little to no dinosaur fossils is always worth publishing. Overall, the manuscript is well written, although there are a few instances of colloquial wording rather than scientific terminology used in describing sedimentary features and structures that should be modified (see detailed comments below). The review of the sedimentological literature is thorough. A few recent articles on dinosaurs from marine deposits and ornithomimids relevant to the discussion have not been consulted (see detailed comments below) in the study and should be done during the round of revisions. The article is well structured and self-contained, no problems on those aspects.

Experimental design

The article is in line with the aims and scope of PeerJ. The research questions are relevant and of interest to the paleontological community as very few fossils have been discovered in the Fox Hills Formation. Identification of the isolated bones is largely based on a literature review, it doesn’t sound like the authors looked at much comparative material first-hand (possibly due to the COVID pandemic) as few specimens number are listed. While this approach is understandable, it can be a limiting factor as the amount of intraspecific variability cannot be accounted for. In instances where the authors state that their specimens are most similar to specific specimens or phalanges (e.g., right phalanx II-1 of Specimen X or phalanx III-2 of Specimen Y), the authors could add a photo of the comparative specimens for the reader to see those similarities. These photos could be added to the current figures without burdening them and making them too complex. Given the age of their fossil material (late Maastrichtian), it is somewhat surprising that the authors did not compare their material to late Maastrichtian large ornithomimids such as Struthiomimus sedens from the Hell Creek Formation or the recently described large ornithomimid from the Scollard Formation of Alberta (Nottrodt 2022 in JVP). Comparison with these large species (and maybe even the slightly older taxon Gallimimus from Mongolia) would be most relevant for this study.

Validity of the findings

Overall, I believe the authors’ interpretations are largely supported by the data presented. Although comparison with other large ornithomimids would be good, I believe the authors are correct in attributing the fossil material to ornithomimids. However, I disagree with the taphonomic interpretation of the Fairpoint phalanx. It looks like the extremities of this specimen have been abraded by water transport and the shaft water-polished prior to burial. The roundness of the extremity does not appear to be solely due to the formation of concretions. Furthermore, interpretation of the depositional environment of the Fairpoint phalanx seems to be inconsistent within the manuscript and with sedimentary structures exposed at the site. In the abstract, the authors state that the phalanx was deposited on a “beachfront or nearshore sandbar” (line 44) whereas there is no detailed interpretation of the setting in the manuscript (lines 404-428). Additionally, the authors describe (lines 121-123) and illustrate (Figure 3A) sedimentary structures present at the site as being hummocky cross-stratification, sedimentary structures well-known to be the result of storm wave-base activity at depth, which contradict the authors’ beachfront and sandbar interpretation. The phalanx definitely comes from a deeper nearshore environment than what they interpreted.

Additional comments

Additional comments
- Lines 50-55: Although the beginning of the abstract mentions that 2 fossils are reported, the mention and description of the second element feels more like an afterthought. Maybe integrate the it better into the abstract and provide the same level of details and information as the first element.
- Line 61 and throughout the manuscript: modifiers (early, late) to stage names are in lower case.
- Line 121: what does “massively bedded” mean? Do you mean that it generally doesn’t display sedimentary structures except for those you list afterwards? If so, phrase it that way. Do so throughout the manuscript.
- Line 130: Is it possible to indicate the location of the fossil lag and theropod phalanx in the outcrops shown in Figure 3A or are they from a different area?
- Line 144: Myledaphus is not a ray but a guitarfish
- Line 147: what is a “channel cut”? A channel, a channel scour, an infilled channel??? Reword with proper terminology. Do so throughout the manuscript.
- Line 154: replace “lies on top of” with “overlies”
- Line 159: replace “massively bedded sands” with “structureless sandstone”
- Line 160: replace “silts and sands” with “siltstones and sandstones”. They are all lithified.
- Lines 193-200: run-on sentence. Break it into shorter sentences.
- line 234: a geologic age “rises”?? Reword.
- Line 263: refer to Figure 5 illustrating the specimen here.
- Line 291: I don’t think the condyle is lost to “diagenesis”, it is lost to pre-burial abrasion. That taphonomic story should be described.
- Line 304: do you mean Lambe 1917 instead of 2017?
- Lines 306-309: but those differences are relative to a very large adult T. rex. These differences are undoubtedly related to body size. How does your phalanx compare to juvenile and subadult T. rex or even with albertosaurines? The large size and expanded articular regions are far less pronounced in those because they don’t reach the size of the largest T. rex known. I know you mention that your specimen could be a juvenile individual later in your description, but maybe point out right away that those differences could be due to ontogeny.
- Line 316: spell out Horseshoe Canyon Formation and reword sentence.
- Line 323-325: state clearly what statement refers to the Horseshoe Canyon tyrannosaurid (called Albertosaurus sarcophagus) and what statement refers to your specimen. I think you do a comparison but it’s unclear.
- Line 340: differences in length are somewhat irrelevant as they are likely due to body size/ontogeny, especially since O velox and the Bissekty ornithomimid are small taxa/individuals. It’s the morphological similarities/differences that you should focus in your comparisons.
- Line 341: what do toy mean by “most consistent”? What makes your specimen most consistent with phalanx III-2 when you mentioned earlier that your specimen shared similiarities with phalanx II-1?
- Line 345: specify the specimen numbers referred to by Cullen et al (2013).
- Lines 347: probably more relevant (in terms of geologic age) to your Fox Hills phalanx, you should do comparisons with time-equivalent ornithomimids, like the large Struthiomimus sedens from the Hell Creek Fm and the recently described large ornithomimid from the Scollard Fm of Alberta (Nottrodt 2022 in JVP).
- Line 362: refer to Figure 6 illustrating the specimen here.
- Line 400: as for the first phalanx, you should do comparisons with the large Struthiomimus sedens from the Hell Creek Fm and the recently described large ornithomimid from the Scollard Fm of Alberta.
- Line 410-428: in your description of the fossil site at the beginning of the manuscript, you mentioned the presence of hummocky cross-stratification. This type of sedimentary structure is a very clear indicator of depositional environment, specifically of storm wave-base at the bottom of a shallow nearshore environment. This needs to be described in details in this section as it directly indicates the environment and depth at which the bone assemblage accumulated.
- Lines 414-418: You have never referred to your fossil assemblage site as White Owl assemblage. You’ve also mentioned on ine 164 that it’s part of the Fairpoint Member, not the White Owl Member. You’ve only referred to your specimen by catalogue number of geographic location or its member (i.e., Fairpoint). You need to refer to the site/specimen with consistency throughout the manuscript.
- Lines 420-423: similar mixed assemblages have been reported in the upper part of the Dinosaur Park Formation of Alberta as well. Those are not uncommon. Need to increase the number of references being cited.
- Line 426: again, the term White Owl bone locality has never been used and it appears to be a misnomer.
- Line 428: when plural, the word “formations” is written with a lowercase.
- Line 431: Here you refer to your theropod phalanx as the Fairpoint phalanx, not the White Owl phalanx.
- Line 435-437: awkward wording. “… the deltaic setting then beginning to dominate the northern parts of the Western Interior.” The “then beginning” is awkward, but also the northern parts of the Western Interior wasn’t dominated by a delta. Maybe in ND and SD, but definitely not farther north where’s the setting is either fully terrestrial or lacustrine.
- Lines 437-438: earlier in the paragraph (line 430) you said that the depositional settings for the two phalanges were “essentially similar” but here you say that one is more onshore than the other. I’m inclined to agree with the latter statement, so rephrase the beginning of the paragraph.
- Line 454: add “NALMA”
- Line 455: just say “Taphonomy of …” rather than “Bone taphonomy…”
- Lines 455-512: the degree of roundness of the extremities of the phalanx and the apparent “sheen” of the surface of the shaft are reminiscent of abrasion (due to transport) and water polish, respectively. This phalanx appears to have been exposed and rolled by water quite a bit before being finally buried rather than the element being buried while soft tissue was still attached to it and pyrite concretions forming on it. This scenario would also be consistent with the orthogonal (i.e., dry bone) fractures and flaking observed on the bone. The concretion could have formed once the phalanx reached its final location.
- Line 568: proper name of the rock unit is the Bearpaw Formation
- Line 571: Drysdale et al. (2020) also described 3 hadrosaur skeletons found in the Bearpaw Formation of Alberta, which should be added to the list. The nodosaur described by Brown et al. (2017) is from the Cenomanian and the therizinosaur described by Zanno et al. (2009) is from the Turonian, so both are from much older phases of the WIS than the other specimens described. This should be mentioned.
- Line 574: technically-speaking, coastal includes “onshore”
- Line 575: that sentence makes it sound like the Fox Hills represent a “large part of the Late Cretaceous” when it is only a part of the Maastrichtian. Need to rephrase.
- Line 584: I believe that it’s a bit of an exaggeration to say that “Late [sic] Maastrichtian marginal Seaway sediment holds considerable promise to…” With just a few isolated teeth and bones at best identifiable to the family level, it’s hard to imagine how useful these specimens can be to “interpret the complex depositional patterns and paleoenvironmental shifts that occurred during the waning of the WIS…” Your discovery is important in its own right, there is no need to over-hype it.

---

## Round 0.2 · accepted · Accept

I have checked the manuscript and response to both reviewers, and am happy that this has all been dealt with very well. I thus judge that the manuscript is ready for publication. Congratulations!